# AnyText2: Visual Text Generation and Editing With Customizable Attributes

## Abstract

With the ongoing development in the text-to-image(T2I) domain, accurately generating text within images seamlessly integrating with the visual content has garnered increasing interest from the research community. In addition to controlling glyphs and positions of text, there is a rising demand for more fine-grained control over text attributes, such as font style and color, while maintaining the realism of the generated images. However, this issue has not yet been sufficiently explored. In this paper, we present **AnyText2**, the first known method to achieve precise control over the attributes of every line of multilingual text when generating images of natural scenes. Our method comprises two main components. First, we introduce an efficient WriteNet+AttnX architecture that encodes text features and injects these intermediate features into the U-Net decoder via learnable attention layers. This design is 19.8% faster than its predecessor, AnyText, and improves the realism of the generated images. Second, we thoroughly explore methods for extracting text fonts and colors from real images, and then develop a Text Embedding Module that employs multiple encoders to separately encode the glyph, position, font, and color of the text. This enables customizable font and color for each text line, yielding a 3.3% and 9.3% increase in text accuracy for Chinese and English, respectively, compared to AnyText. Furthermore, we validate the use of long captions, which enhances prompt-following and image realism without sacrificing text writing accuracy. Through comprehensive experiments, we demonstrate the state-of-the-art performance of our method. The code and model will be open-sourced in the future to promote the development of text generation technology.

## 1 Introduction

Diffusion-based generative models Ho et al. (2020); Rombach et al. (2022); Ramesh et al. (2021; 2022); Podell et al. (2024) have gained prominence due to their ability to generate highly realistic images with intricate details, and gradually replacing previous technologies like GANs Goodfellow et al. (2014) and VAEs Kingma & Welling (2014). In recent research, models such as DALL·E3 OpenAI (2023), Stable Diffusion 3 Esser et al. (2024), and FLUX.1 BlackForestLab (2024) have enhanced their visual text rendering capabilities through the introduction of new technologies, such as encoding image captions using large language models like T5, or employing rectified flow transformers. However, their performance of state-of-the-art models in text rendering still falls short of expectations. Therefore, many researchers aim to inject or enhance text rendering capabilities into pre-trained diffusion models using various technical methods, while maintaining their diversity and realism in image synthesis. These methods, such as GlyphDraw Ma et al. (2023), GlyphControl Yang et al. (2023), TextDiffuser Chen et al. (2023b), AnyText Tuo et al. (2023), TextDiffuser-2 Chen et al. (2023a), Glyph-SDXL Liu et al. (2024a), Glyph-SDXL-v2 Liu et al. (2024b), GlyphDraw2 Ma et al. (2024), not only significantly improves the accuracy of text rendering but also extends functionalities such as multilingual text generation, text editing, automatic or specified layout, and even customizable text attributes.

There are generally two mechanisms for injecting text rendering capabilities into pre-trained models: (1) *conditional embeddings* in the prompt and (2) *auxiliary pixels* in the latent space. The first approach encodes the visual appearance of each character as embeddings and combines them with the image caption to serve as conditions; notable methods include TextDiffuser-2, Glyph-SDXL, and

Glyph-SDXL-v2. The second approach involves using character-level segmentation mask or pre-rendered glyph images and injects into the latent space as catalysts for text rendering; representative methods include TextDiffuser and GlyphControl. While conditional embeddings require a relatively large amount of training data to encode the visual appearance of various styles for each character, they struggle with generalization for unseen characters. In contrast, although auxiliary pixels can leverage the visual appearance of pre-rendered characters off-the-shelf, the resulting text accuracy and integration within the image are generally poor due to the absence of text-related information in the image caption. To address these limitations, approaches like AnyText and GlyphDraw2 adopt a combined strategy. Our proposed method, AnyText2, follows a similar approach but diverges by not encoding auxiliary pixels and conditional embeddings in a ControlNet-like manner, as it not only controls text rendering at each time step but also participates in generating image content, which is inefficient and detrimental to image quality. Instead, we designed the WriteNet module, focusing solely on text rendering. In this method, text-related features are encoded just once and then fused with image content at each time step through learnable AttnX layers inserted into the U-Net decoder. This streamlined approach significantly improves inference speed while enhancing image realism.

Most methods primarily inject only glyph information into the conditional embeddings. However, the token embeddings cannot be directly associated with the corresponding areas in the image. Our research indicates that further encoding the positional information of each line into the tokens can significantly enhance text accuracy. Additionally, we integrate font style and color information through dedicated encoders, incorporating them into the tokens. This not only improves accuracy but also facilitates attribute customization. Previous methods, such as DiffSTE Ji et al. (2023) and Glyph-SDXL Liu et al. (2024a), have also achieved text attribute customization, but they typically rely on synthetic images for training. This reliance stems from the difficulty of extracting font and color labels from natural scene images. However, it poses two significant drawbacks. First, since fonts and colors are described textually in image captions, any font or color name not present in the training dataset is ineffective. In contrast, our method encodes font styles directly from a text line image, which can either be rendered using a user-specified font file or selected from another image using a brush tool. Regarding color, our method enables users to specify RGB values directly through a color picker or palette, eliminating reliance on vague color names. Second, training on synthetic data primarily results in generating overlaid text that applicable in scenarios like posters and cards. However, such text can often be produced using existing image editing software, which not only ensures text accuracy but also allows for perfect specification of text font and color. To our knowledge, our method is the first to enable customized text attribute generation in open-domain scenarios (e.g., foods, products, signboards). By extracting font styles and colors from images and utilizing specially designed encoders for feature encoding, we generate both overlaid and embedded text applicable in any context. Selected examples are presented in Fig. 1.

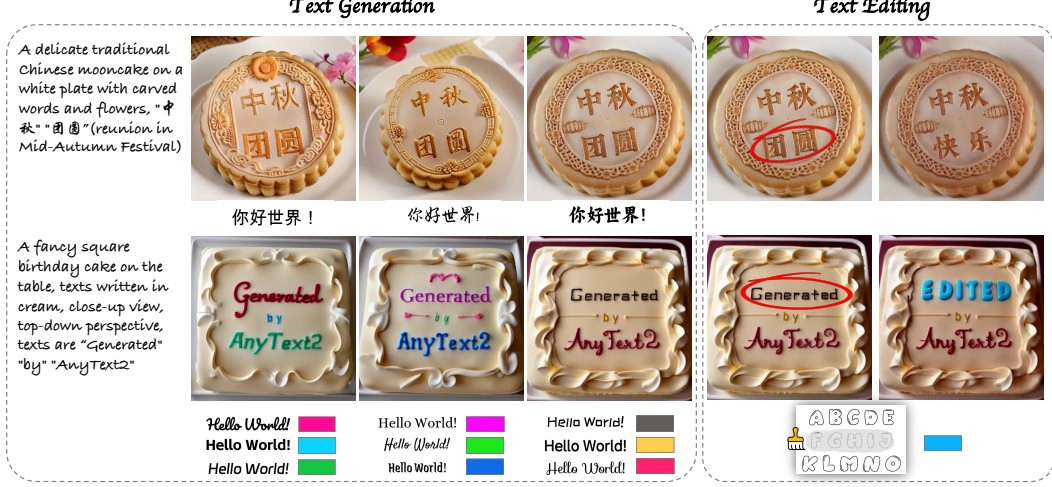

Figure 1: AnyText2 can accurately generate multilingual text within images and achieving a realistic integration. Furthermore, it allows for customize attributes for each line, such as controlling the font style through font files or mimic from an image using a brush tool, and specifying the text color. Additionally, AnyText2 enables customizable attribute editing of text within images.

## 2 RELATED WORKS

**Controllable Text-To-Image Generation** In T2I models, achieving precise control through pure textual descriptions poses significant challenges, and a multitude of methods have emerged. Among the pioneering works are ControlNet Zhang & Agrawala (2023), T2I Adapter Mou et al. (2023), and Composer Huang et al. (2023), leverage control conditions such as depth maps, pose images, and sketches to guide image generation. Another category is comprised of subject-driven methods, such as Textual Inversion Gal et al. (2023), DreamBooth Ruiz et al. (2022), IP-Adapter Ye et al. (2023), ReferenceNet Hu et al. (2023), InstantID Wang et al. (2024), and PhotoMaker Li et al. (2024). These methods focus on learning representation of specific subject or concept from one or a few images, primarily ensuring identity preservation in the generated images while allowing less stringent control over other attributes such as position, size, and orientation. Visual text generation can be viewed as a sub-task within this framework if we consider each character as an identity. Our goal is to control the positions and strokes of characters, but without the rigid constraints characteristic of methods like ControlNet. Instead, we seek to introduce some diversity in font style, size, and orientation, while ensuring that the characters remain legible.

**Visual Text Generation** The text encoder plays a crucial role in generating accurate visual text, as highlighted by Liu et al. (2023). Many subsequent methods adopted character-level text encoders to incorporate word spelling or character visual appearance into conditional embeddings, such as DiffSTE Ji et al. (2023), TextDiffuser-2 Chen et al. (2023a), UDiffText Zhao & Lian (2023), and Glyph-SDXL Liu et al. (2024a). However, training a text encoder independently requires significant resources and limits the range of writable characters. For instance, while Glyph-SDXL-v2 expanded its text encoder Glyph-ByT5 to multilingual, each language is confined to a fixed set of common characters. Conversely, methods like TextDiffuser Chen et al. (2023b) and GlyphControl Yang et al. (2023) utilize character masks or pre-rendered glyph images to assist in text generation. However, the quality of generated images is often subpar because the image caption lacks any text-related information. AnyText Tuo et al. (2023) addresses this by using pre-rendered glyph images as auxiliary pixels and employing a pre-trained OCR model Li et al. (2022) to encode strokes, which are then integrated into the conditional embeddings. The latest GlyphDraw2 Ma et al. (2024) adopts a similar approach. Our proposed method also follows this strategy but utilizes an innovative architecture, WriteNet+AttnX, to achieve a more efficient and effective fusion of image and text. Additionally, methods like UDiffText, Brush Your Text Zhang et al. (2023), and Glyph-SDXL impose restrictive interventions on attention maps in corresponding text areas to improve accuracy. In contrast, our approach employs a position encoder to encode text position and injects it into the conditional embeddings, allowing for spatial awareness.

**Text Attributes Customization**

Currently, there are numerous studies on font style transfer based on GANs or diffusion models, including LF-Font Park et al. (2021a), MX-Font Park et al. (2021b), Diff-Font He et al. (2024), and FontDiffuser Yang et al. (2024). These endeavors, categorized as Few-shot Font Generation (FFG), focus on learning font style representation from one or a few reference images and transforming the input source image into a target image that closely matches that style. While our task shares similarities that involve decoupling content and style from reference characters, our objective is to generate text in a specified style directly onto an image, contrasting with their goal of automatically creating a font library made up of plain characters. As for color control, current T2I models typically struggle to interpret the original RGB values provided in prompts. To address this issue, some works focus on color prompt learning. For instance, ColorPeel Butt et al. (2024) constructs a synthetic dataset and decouples color and shape during training, allowing for the learning of specific color tokens and using them to achieve precise color control. In the realm of visual text generation, leveraging synthetic data for text attribute customization is also an intuitive approach. Notably, DiffSTE Ji et al. (2023) and Glyph-SDXL Liu et al. (2024a) incorporate the relevant font type names and color names directly into the prompts, enabling the diffusion model to learn the concepts linked to these specific names. This facilitates precise control over text attributes during inference.

## 3 METHODOLOGY

Most T2I models excel at generating diverse and realistic images but have limitations in text generation capabilities. AnyText2 is designed as a plugin for these models, processing text signals

separately and injecting them into the T2I models. The framework of the proposed AnyText2 is depicted in Fig. 2.

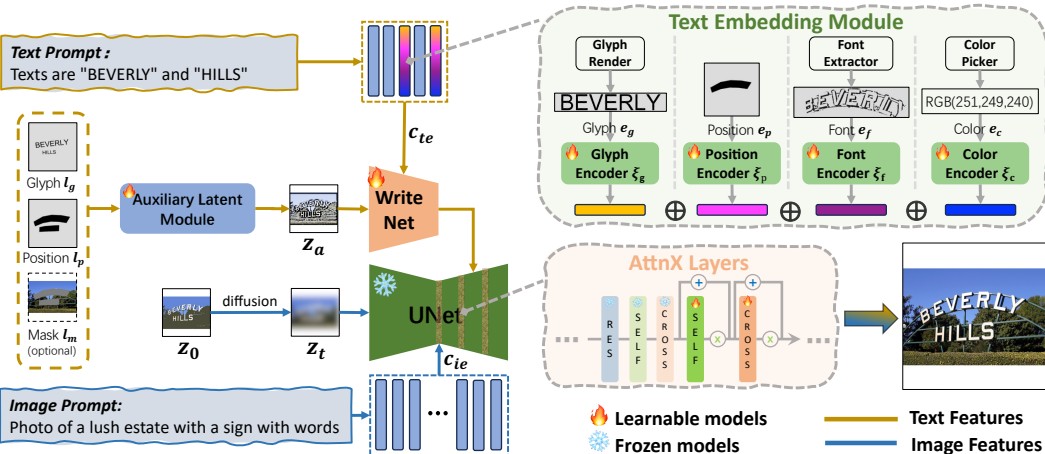

Figure 2: The framework of AnyText2, which is designed with a WriteNet+AttnX architecture to integrate text generation capability into pre-train diffusion models, and there is a Text Embedding Module to provide various conditional control for text generation.

In the standard Latent Diffusion Model (LDM) Rombach et al. (2022), the original latent pixels $z_0$ are gradually adding noise $\epsilon$ through a forward diffusion process to obtain a noisy latent pixels $z_t$. The image prompt is then encoded into conditional embeddings $c_{ie}$ using a pre-trained CLIP Radford et al. (2021) text encoder. Both $z_t$ and $c_{ie}$ are then fed into a conditional U-Net Ronneberger et al. (2015) denoiser $\epsilon_\theta$ to predict the noise. The final image is generated after $t$ time steps of the reverse denoising process. To enhance text generation capabilities, we introduce an Auxiliary Latent Module that encodes the glyph, position, and optionally a masked image (to enable text editing), producing auxiliary pixels $z_a$. The text prompt is processed through a Text Embedding Module to obtain the conditional embeddings $c_{te}$. This Module comprises multiple encoders designed to facilitate various conditional controls. Both $(z_t, c_{ie})$ and $(z_a, c_{te})$ undergo cross-attention computations in U-Net and WriteNet to better guide the image and text generation, respectively. The integration of image and text is then performed through the U-Net decoder with inserted AttnX layers. More formally, the optimization objective of our method is represented by the following equation:

$$\mathcal{L} = \mathbb{E}_{z_0, c_{ie}, z_a, c_{te}, t, \epsilon \sim \mathcal{N}(0,1)} \left[ \| \epsilon - \epsilon_\theta(z_t, z_a, c_{te}, c_{ie}, t) \|_2^2 \right] \quad (1)$$

Next, we will provide a detailed introduction to the WriteNet+AttnX architecture and the Text Embedding Module.

## 3.1 WRITENET+ATTNX

We analyze the principle underlying AnyText in Appendix A. It utilizes a ControlNet-like module, TextControlNet, which is responsible not only for encoding text information but also for generate image content in collaboration with the U-Net. While this integration facilitates a seamless blending of text and image, it presents two drawbacks: first, the training data for text generation often contains a substantial amount of low-quality images with chaotic text, potentially reducing overall image quality. Second, it necessitates computation at each time step, thereby lowering inference efficiency. Thus, we decouple text and image generation, while the production-ready U-Net that trained with billions of images is responsible solely for generating image content. As for WriteNet, drawing the insights from ControlNet, we clone a trainable copy from the U-Net encoder and connect it to the U-Net decoder via zero convolution. However, to concentrate exclusively on learning how to write text, we remove the timestep layers and any image-related inputs, such as the noisy latent $z_t$, and descriptions of image content in the prompt.

We find that directly connecting the intermediate features output by WriteNet to the frozen U-Net decoder does not yield seamless blending with image content. To address this, auxiliary pixels

must adequately interact with image latent pixels and conditional embeddings. This interaction can be facilitated through the self-attention and cross-attention in each attention block. Therefore, we insert a self-attention and a cross-attention layer, denoted as AttnX layers, in every attention blocks in the middle and decoder part of the U-Net. The parameters of these layers are copied from the corresponding layers of the current block and set to be trainable. Since the U-Net decoder is skip-connected with the encoder, which directly receives the auxiliary pixels $z_a$, the trainable AttnX near the output can inadvertently copy text glyphs from $z_a$, potentially degrading the fusion effect (further details in Sec. 5.3.1). To mitigate this, we only insert these layers in the first two blocks of the U-Net decoder, specifically at resolutions of 16x16 and 32x32. The output from each AttnX layer is multiplied by a strength coefficient $\alpha$ and combined with the output from the previous layer through a shortcut connection. By adjusting $\alpha$, we can modulate the fusion strength between text and image, as illustrated in Fig. 3. Notably, setting $\alpha = 0$ and multiply the WriteNet output by 0 enables AnyText2 to generate images without text, solely utilizing the original diffusion model.

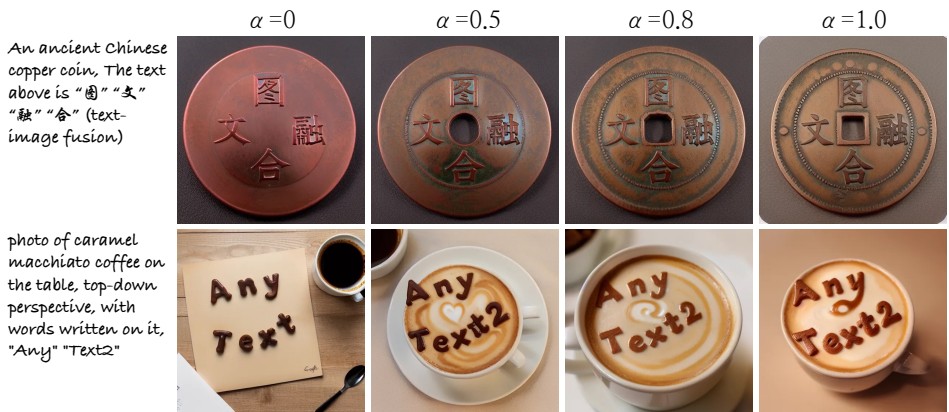

Figure 3: By adjusting the strength coefficient $\alpha$ from 0 to 1, shows that the text-image fusion is gradually improving.

## 3.2 TEXT EMBEDDING MODULE

In AnyText, each line of text is rendered onto an image denoted as $e_g$ by a glyph render. The glyph information is then encoded using the glyph encoder $\xi_g$, which comprises an OCR model and a linear layer. We build upon this approach by incorporating three additional encoders: $\xi_p$, $\xi_f$, and $\xi_c$. These encoders serve to encode the position image $e_p$, font image $e_f$, and text color $e_c$, respectively. The output embeddings from these encoders are then summed to produce a representation $r_i$, which effectively captures the attributes of the i-th text line:

$$r_i = \xi_g(e_g) + \xi_p(e_p) + \xi_f(e_f) + \xi_c(e_c) \qquad (2)$$

For the text prompt $y_t$, each text line is replaced with a special placeholder $S_*$. After performing tokenization and embedding lookup, denoted as $\phi(\cdot)$, embeddings of all token are obtained. We then substitute back the attribute representation of $n$ text lines at $S_*$, and use CLIP text encoder $\tau_\theta$ to generate the final conditional embeddings $c_{te}$:

$$\boldsymbol{c}_{te} = \tau_\theta(\phi(y_t), r_0, r_1, ..., r_{n-1}) \qquad (3)$$

Next, we will provide a detailed introduction to the position, font, and color encoders.

### 3.2.1 POSITION ENCODER

In the Cross-Attention layers, the flattened auxiliary pixels $\varphi(z_a)$ are projected into a query matrix $Q = W_Q \cdot \varphi(z_a)$, the conditional embeddings are projected into a key matrix $K = W_K \cdot \varphi(c_{te})$ and a value matrix $V = W_V \cdot \varphi(c_{te})$, via learned linear projections $W_Q, W_K, W_V$. The attention map is computed as:

$$\mathcal{M} = Softmax(\frac{QK^T}{\sqrt{d}}) \qquad (4)$$

where $d$ is the projection dimension. the cross-attention output $\mathcal{M} \cdot V$ is used to update visual features. Intuitively, $\mathcal{M}$ reflects the similarity between Q and K, and the element $\mathcal{M}_{ij}$ defines the weight of the $j$-th conditional embedding on the $i$-th auxiliary pixel. However, the embedding of a particular text line is not explicitly associated with the pixels corresponding to its text area. Therefore, we introduce a position encoder $\xi_p$ that employs four stacked convolutional layers to encode the position image $e_p$, followed by an average pooling layer to adjust the shape and add it to the original embedding. By utilizing $\xi_p$, we introduce spatial information for each text line, enabling the embedding to achieve spatial awareness. In the ablation study in Sec. 5.3.1, we demonstrate that this significantly improves the accuracy of text generation.

### 3.2.2 FONT ENCODER

Extracting text from natural scene images is quite challenging due to complex lighting variations and background interference. Instead of striving for precise separation from the complex background, we use a straightforward adaptive threshold on the text regions to construct a font extractor, resulting in a rough binary image that serves as the font image $e_f$. To prevent font style leakage in glyph image $l_g$ and $e_g$, each text line is rendered using a random font. Similarly, to mitigate glyph leakage from $e_f$, various transformations such as rotation, translation, scaling, and occlusion are applied in the font extractor. Examples of the obtained font image $e_f$ can be found in Appendix B. Due to noise in font images, many network architectures, such as stacked convolutional layers or the pre-trained DINOv2 Oquab et al. (2023), struggle to effectively encode font features. Here, we employ an OCR model combined with a linear layer to construct our font encoder $e_f$, as the OCR model inherently focuses on the text portions despite a noisy background. On the other hand, the OCR model naturally perceives font types and is trained to be invariant to font variations, meaning that it produces the same output for text lines with different font types. Therefore, to shift its attention from glyphs to fonts, we allow the parameters to be trainable throughout the training process. We also randomly nullify a certain proportion of $e_f$ during training, to ensure the model can generate text when no font style is specified. During inference, we can either render the text using a user-specified font onto the image or select a text region from an image and input to the font extractor to construct $e_f$. We then utilize the proposed font encoder $\xi_f$ to encode the font style. More examples are illustrated in Fig. 4. Notably, incorporating font style features into the conditional embeddings enhances the similarity between Q and K in Equ. 4, which in turn improves the text accuracy, as detailed in Sec. 5.3.1.

### 3.2.3 COLOR ENCODER

We employ a non-learning method to create a color picker for obtaining the RGB values of the text. Initially, the colors of all pixels within the text region are clustered and ranked, from which we select the top dominant color blocks. We then analyze their shapes and positions using morphological analysis techniques to identify the most likely text blocks, outputting the mean RGB value as the text color $e_c$. According to our statistics, approximately 65% text lines in the training data conform to specific criteria, yielding reliable color labels with an accuracy exceeding 90%. Examples can be found in Appendix B. If a text line does not receive a reliable color label, we assign it RGB(500, 500, 500) as a default, which tends to result in a random color assignment during inference. In constructing the color encoder $\xi_c$, we experimented with various techniques, including Fourier feature encoding and convolutional layers. However, we experimented that a simple linear projection layer was sufficient to encode the three RGB values. More examples are illustrated in Fig. 4.

## 4 DATASET AND BENCHMARK

We utilize AnyWord-3M, a large-scale multilingual dataset proposed by AnyText Tuo et al. (2023) as our training dataset. The AnyWord-3M dataset contains 3.53 million images, representing a diverse array of scenes containing text, such as street views, book covers, advertisements, posters, and movie frames. However, the captions in AnyWord-3M were generated by BLIP-2 Li et al. (2023), which lack detailed and accurate descriptions. To improve this, we regenerated the captions using QWen-VL Bai et al. (2023). Statistics analysis reveals that the BLIP-2 captions contains only 8 words at average, while those generated by QWen-VL is around 47 words, with roughly one-third exceeding 50 words. This substantial increase in caption length significantly enhances the description of image

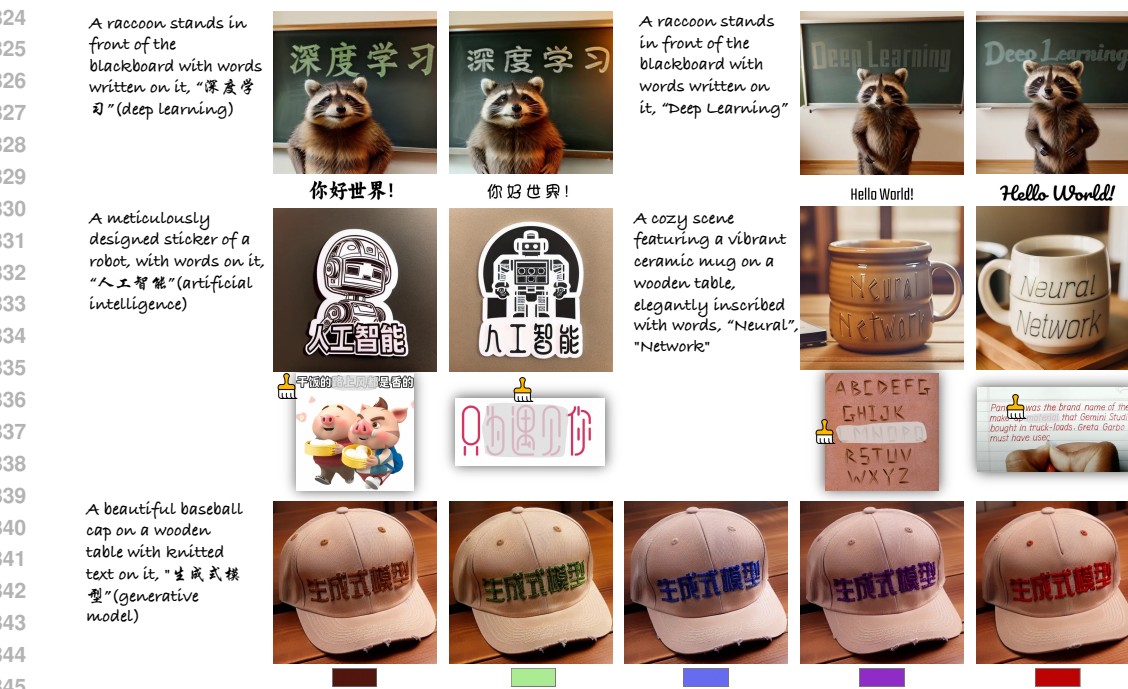

Figure 4: Examples of customizing text attributes. The first row demonstrates font style control using a user-specified font file. The second row showcases selecting a text region from an image to mimic its font style. The third row illustrates the control of text color.

details. In the ablation study detailed in Sec. 5.3.2, we found that while longer captions slightly reduce text accuracy, they significantly improve the model's prompt-following ability. Thus, we opted to train with the longer captions. Examples of training images alongside corresponding long and short captions can be found in Appendix E.

We use the AnyText-benchmark to evaluate the performance of the model, which includes 1,000 images extracted from Wukong Gu et al. (2022) and 1000 images from LAION-400M Schuhmann et al. (2021). This benchmark quantitatively assesses the model's performance in Chinese and English generation, respectively. The AnyText-benchmark employs three evaluation metrics: Sentence Accuracy (Sen. ACC) and Normalized Edit Distance (NED) for measuring text accuracy using the DuGangOCR ModelScope (2023) model, as well as the Frechet Inception Distance (FID) for assessing image authenticity. In addition to these, we incorporate CLIPScore Hessel et al. (2021) to evaluate the model's prompt-following capability.

# 5 EXPERIMENTS

## 5.1 IMPLEMENTATION DETAILS

Our training framework is implemented based on AnyText[1], with model weights initialized from SD1.5[2]. Compared to AnyText, we have only increased the parameters by 4.5%(63.8M), while the design of WriteNet+AttnX architecture has improved the inference speed by 19.8%, as detailed in Appendix D. Unlike AnyText's multi-stage training regimen, AnyText2 adopts end-to-end training. The model was trained for 10 epochs on AnyWord-3M using 8 Tesla A100 GPUs, taking approximately two weeks. We employed the AdamW optimizer with a learning rate of 2e-5 and a batch size of 48. The designs of the Auxiliary Latent Module and glyph encoder are consistent with those in AnyText. The resolutions of $l_g$, $l_p$, $l_m$, and $e_p$ are 512x512, while the resolutions of $e_g$ and $e_f$ are 80x512. The fusion strength coefficient $\alpha$ is configured to 1.0. A probability of 50% is applied

---

[1] https://github.com/tyxsspa/AnyText

[2] https://huggingface.co/runwayml/stable-diffusion-v1-5

Table 1: Quantitative comparison of AnyText2 and competing methods. †is trained on LAION-Glyph-10M, and ‡is fine-tuned on TextCaps-5k. Numbers in brown color represent the results obtained using the long caption version of the AnyText-benchmark.

| Methods | English | | | | Chinese | | | |
|---|---|---|---|---|---|---|---|---|
| | Sen.ACC↑ | NED↑ | FID↓ | CLIPScore↑ | Sen.ACC↑ | NED↑ | FID↓ | CLIPScore↑ |
| ControlNet | 0.5837 | 0.8015 | 45.41 | 0.8448 | 0.3620 | 0.6227 | 41.86 | 0.7801 |
| TextDiffuser | 0.5921 | 0.7951 | 41.31 | 0.8685 | 0.0605 | 0.1262 | 53.37 | 0.7675 |
| GlyphControl† | 0.3710 | 0.6680 | 37.84 | 0.8847 | 0.0327 | 0.0845 | 34.36 | 0.8048 |
| GlyphControl‡ | 0.5262 | 0.7529 | 43.10 | 0.8548 | 0.0454 | 0.1017 | 49.51 | 0.7863 |
| Anytext | 0.7239 | 0.8760 | 33.54 | 0.8841 | 0.6923 | 0.8396 | 31.58 | 0.8015 |
| | 0.7242 | 0.8780 | 35.27 | 0.9602 | 0.6917 | 0.8373 | 31.38 | 0.8870 |
| GlyphDraw2 | 0.7369 | 0.8921 | - | - | - | - | - | - |
| Anytext2 | **0.8096** | **0.9184** | **33.32** | **0.8963** | **0.7130** | **0.8516** | **27.94** | **0.8139** |
| | **0.8175** | **0.9193** | **27.87** | **0.9882** | **0.7250** | **0.8529** | **24.32** | **0.9137** |

to choose between inputting $l_m$ or an empty image, facilitating training for both text generation and editing. A probability of 20% is used to input an empty $e_f$, enabling the model to generate random fonts when no font style is specified. Approximately 35% of $e_c$ are assigned a default value due to the absence of accurate color labels, allowing for the generation of text in random colors when no color is specified.

## 5.2 COMPARISON RESULTS

### 5.2.1 QUANTITATIVE RESULTS

AnyText2 excels not only in generating accurate text but also in text editing, attribute customization, and effective prompt-following in images. Our subsequent ablation study confirmed that some of these features may slightly reduce text accuracy. Nevertheless, AnyText2 significantly outperforms all competing methods in terms of accuracy while maintaining superior image realism and prompt-following capabilities. We evaluated ControlNet Zhang & Agrawala (2023), TextDiffuser Chen et al. (2023b), GlyphControl Yang et al. (2023), AnyText Tuo et al. (2023), and GlyphDraw2 Ma et al. (2024) using the benchmarks and metrics outlined in Sec. 4. To ensure a fair evaluation, all publicly available methods employed the DDIM sampler with 20 sampling steps, a CFG scale of 9, a fixed random seed of 100, a batch size of 4, and consistent positive and negative prompt words. The quantitative comparison results are presented in Table 1. For GlyphDraw2, we referenced the metrics reported in their paper, achieving a 7.27% improvement in English Sentence Accuracy (Sen. ACC). A comparison for Chinese was not included, as they utilized the PWAcc metric and excluded some English images during evaluation, and insufficient details were provided. Notably, AnyText2 outperformed AnyText across all evaluation metrics, particularly in the long caption scenario, where it improved English and Chinese Sen. ACC by 9.3% and 3.3%, respectively. Furthermore, it demonstrated significant enhancements in image realism (FID) and prompt-following (CLIPScore).

### 5.2.2 QUALITATIVE RESULTS

As Shown in Fig. 5, we conducted a qualitative comparison of AnyText2 with several recent methods, including TextDiffuser-2 Chen et al. (2023a), Glyph-SDXL-v2 Liu et al. (2024b), Stable Diffusion 3 Esser et al. (2024), and Flux.1 BlackForestLab (2024). The image captions in the leftmost column are input directly to TextDiffuser-2, SD3, and FLUX.1. For Glyph-SDXL-v2 and AnyText2, the inputs were adjusted according to each method's requirements, such as manually setting the layout, selecting appropriate text fonts or colors according to the captions. Each method underwent multiple trials and one of the best is presented. From the results, TextDiffuser-2 shows subpar performance in text accuracy, especially when handling multiple lines. Glyph-SDXL-v2 achieves good accuracy and enables precise customization of text fonts and colors, and it is the only competitive method that supports multilingual generation, but it can only generate overlaid text on images, and the generated text appears to have no correlation with the image content. SD3 provides visually appealing images but its English accuracy is moderate, with only rough control over colors and almost no control over fonts. FLUX.1, as the leading text-to-image model currently available, produces impressive visual results while maintaining decent English accuracy, albeit with occasional capitalization errors. It permits rough control over simple fonts and colors in the prompt but is limited to

English generation. In comparison, AnyText2 stands out with the best accuracy, the most precise control over text attributes, seamless text-image integration, and multilingual support.

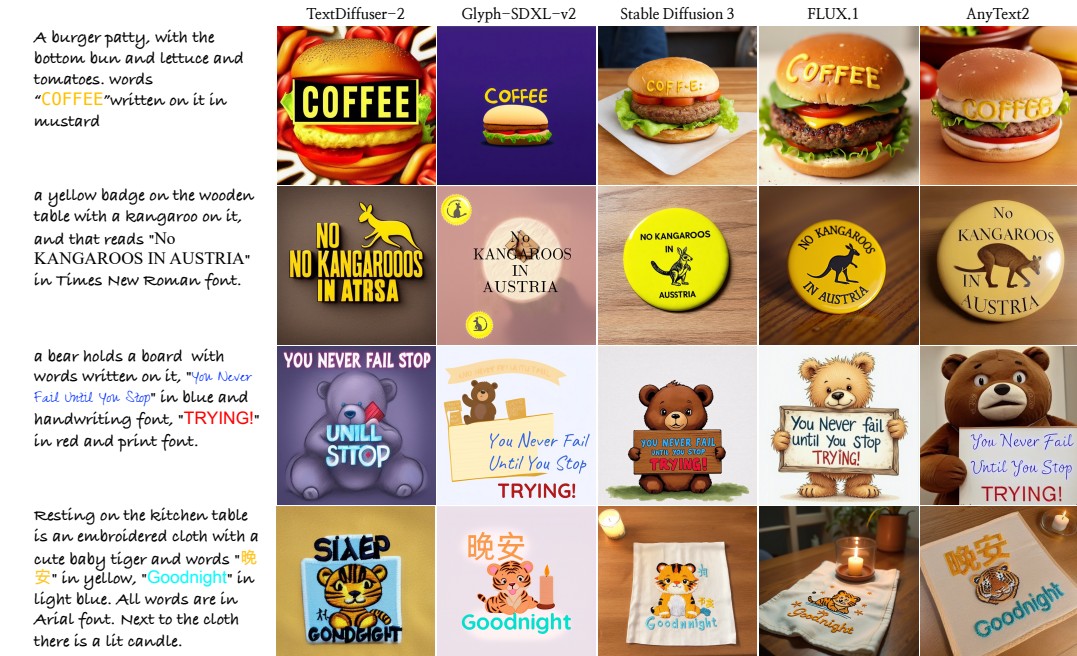

Figure 5: Qualitative comparison of AnyText2 and competing methods. From the perspectives of text accuracy, text-image integration, attribute customization, and multilingual support, AnyText2 demonstrated significant advantages.

## 5.3 ABLATION STUDY

Following AnyText, we extracted 200k images from AnyWord-3M, which includes 100k images each for Chinese and English. We conducted ablation experiments by training on this small-scale dataset for 15 epochs to validate each module of our method. Next, we will analyze our method from two perspectives: accuracy and realism, as well as prompt-following capability.

### 5.3.1 ACCURACY AND REALISM

In Table 2, we validated the effectiveness of each module in AnyText2. Specifically, in Exp.1, the original AnyText serves as the baseline. By incrementally adding the position and font encoders in the Text Embedding Module in Exp.2&3, there is a significant boost in text accuracy. This improvement is attributed to the enhanced similarity between auxiliary pixels and conditional embeddings, as analyzed in Sec. 3.2. In Exp.4, adding the color encoder caused a slight decline in text accuracy. We speculate that this may be due to a considerable proportion of incorrect ground truths in the color labels and the challenges of having the model learn the colors of text strokes against complex backgrounds. In Exp.5, 6, and 7, we experimentally demonstrated that the AttnX layers further improve text accuracy; however, their position significantly impacts the FID score. Specifically, the closer the AttnX layer is to the output layer of the U-Net decoder, the more it tends to learn glyphs from the encoder's auxiliary pixels and overlays them on the image due to the skip connections in U-Net. Considering both accuracy and realism, we chose to insert AttnX into the first two blocks of the U-Net decoder, as done in Exp.6. Additionally, in Exp.8, we replaced the ControlNet-like module with WriteNet. Although this led to a slight decrease in accuracy, it significantly improved the FID score. This aligns with our expectations, as there can be a trade-off between image realism and text accuracy; embedded text is often more challenging for OCR to recognize compared to overlaid text, despite offering greater realism. Moreover, WriteNet effectively reduces computational overhead. Considering aspects such as accuracy, realism, and inference efficiency, we opted for the configuration used in Exp.8 for training on the full dataset.

Table 2: Ablation experiments of AnyText2 on a small-scale dataset from AnyWord-3M. The results validate the effectiveness of each submodule in AnyText2.

| Exp. | Pos. | Font | Color | AttnX | | | WriteNet | English | | | Chinese | | |
|---|---|---|---|---|---|---|---|---|---|---|---|---|---|
| | | | | 2 | 1 | 0 | | Sen.ACC↑ | NED↑ | FID↓ | Sen.ACC↑ | NED↑ | FID↓ |
| 1 | | | | | | | | 0.4873 | 0.7721 | 35.38 | 0.5404 | 0.7631 | 31.19 |
| 2 | ✓ | | | | | | | 0.5237 | 0.7876 | 35.86 | 0.5681 | 0.7725 | 29.45 |
| 3 | ✓ | ✓ | | | | | | 0.5926 | 0.8276 | 38.57 | 0.5688 | 0.7763 | 31.60 |
| 4 | ✓ | ✓ | ✓ | | | | | 0.5732 | 0.8169 | 37.24 | 0.5525 | 0.7620 | 32.69 |
| 5 | ✓ | ✓ | ✓ | ✓ | ✓ | ✓ | | 0.6372 | 0.8481 | 44.98 | 0.5632 | 0.7769 | 35.84 |
| 6 | ✓ | ✓ | ✓ | ✓ | ✓ | | | **0.6391** | **0.8490** | 39.14 | **0.5760** | **0.7858** | 32.92 |
| 7 | ✓ | ✓ | ✓ | ✓ | | | | 0.6343 | 0.8478 | 37.21 | 0.5527 | 0.7696 | **28.91** |
| 8 | ✓ | ✓ | ✓ | ✓ | ✓ | | ✓ | 0.6335 | 0.8443 | **35.19** | 0.5614 | 0.7731 | 29.40 |

Table 3: Ablation experiments of training AnyText2 in configuration of Exp.6 using both short(6S) and long captions(6L), and evaluation using the AnyText-benchmark under both short and long caption (marked in brown) scenarios.

| Exp. | Epochs | English | | | Chinese | | |
|---|---|---|---|---|---|---|---|
| | | Sen.ACC↑ | NED↑ | CLIPScore↑ | Sen.ACC↑ | NED↑ | CLIPScore↑ |
| 6S | 15 | 0.6391 | 0.8490 | 0.8797 | 0.5760 | 0.7858 | 0.7941 |
| 6L | | 0.6094 | 0.8296 | 0.8734 | 0.4995 | 0.7401 | 0.7952 |
| 6S | 19 | 0.6313 | 0.8459 | 0.8828 | 0.5719 | 0.7830 | 0.7948 |
| 6L | | 0.6182 | 0.8360 | 0.8773 | 0.5541 | 0.7710 | 0.8036 |
| 6S | 15 | 0.6479 | 0.8481 | 0.8577 | 0.5606 | 0.7738 | 0.7333 |
| 6L | | 0.6305 | 0.8412 | 0.8650 | 0.5055 | 0.7422 | 0.7511 |
| 6S | 19 | 0.6453 | 0.8476 | 0.8596 | 0.5639 | 0.7784 | 0.7372 |
| 6L | | 0.6357 | 0.8431 | 0.8665 | 0.5618 | 0.7738 | 0.7542 |

### 5.3.2 PROMPT-FOLLOWING

We trained two models in the configuration of Exp.6 using short(6S) and long(6L) captions, respectively, to examine the impact of caption length on accuracy and prompt-following. The results are presented in Table 3. The first two rows reveal a noticeable decrease in accuracy when using long captions, particularly in the Chinese Sen. ACC, which dropped by 7.6%. We determined that this decline was partly due to the model using long captions not fully converging on the small-scale training set. Consequently, we continued training for an additional 4 epochs and observed that the metrics for Exp.6S had reached saturation, while those for Exp.6L continued to improve. Though the performance gap between the two models narrowed, the CLIPScore of Exp.6L remained comparable. Next, we replaced the AnyText-benchmark with long captions to evaluate both models and observed a similar trend. After training for 19 epochs, the accuracy gap between the two models further diminished, but Exp.6L's CLIPScore was significantly higher than that of Exp.6S. From these findings, we conclude that training with long captions may cause a slight decrease in text accuracy but enhances prompt-following capabilities, especially for complex captions. Therefore, we decided to use long captions for training on the full dataset.

## 6 CONCLUSION

In this paper, we introduced AnyText2, a novel method that tackles the cutting-edge challenge of precisely controlling text attributes in realistic image generation. We explored techniques for extracting font and color labels from natural scene images and developed dedicated encoders for feature representation, enabling the customization of text attributes for each line. Additionally, we conducted an in-depth analysis of visual text generation mechanisms and creatively proposed the WriteNet+AttnX architecture, which decouples text and image generation tasks while effectively integrating them through attention layers. Our approach outperformed its predecessor, AnyText, achieving higher accuracy, enhanced realism, and faster inference speed. Furthermore, the model's prompt-following capabilities were bolstered through the use of long captions. In future work, we will continue to push the boundaries of visual text generation and aim to gradually port AnyText2 to more innovative diffusion models.

## 7 REPRODUCIBILITY STATEMENT

To ensure reproducibility, we have made the following efforts: (1) We provide implementation details in Sec. 3, Sec. 5.1, Appendix B, and Appendix D, including network structures, intermediate results, training process and selection of hyper-parameters. (2) We provide details on dataset preparation and evaluation metric in Sec 4. (3).We validate the effectiveness of each module on a small-scall dataset in Sec. 5.3. (4) We will release our code and model.

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

## A   ANALYSIS OF THE TEXT GENERATION PROCESS IN ANYTEXT

In this section, we examine the text generation process in AnyText Tuo et al. (2023) and visualize the attention maps of its various cross attention layers using the method from Prompt-To-Prompt Hertz et al. (2023), as shown in Fig. 6. AnyText introduces a Text Embedding Module that extracts text glyph using an OCR model and then fused with other image tokens, and processed through a ControlNet-like network for text generation. We visualized the attention maps of the text tokens in both U-Net and TextControlNet at three resolutions: 64x64, 32x32, and 16x16.

In the UNet encoder (①-③), the process focuses on generating image content without text, then in the TextControlNet (④-⑥), it concentrates on generating text glyphs. Finally, in the UNet decoder

(⑦-⑨), the integration of image and text is achieved. However, as shown in the lower part of Fig. 6, we discovered that TextControlNet also responds significantly to non-text tokens. This indicates that it not only facilitates text generation but also acts as part of the denoiser, working in conjunction with U-Net to generate the overall image content. While this improves the integration of image and text, it can also lead to drawbacks such as decreased overall image quality and decreased inference efficiency. This paper proposes the WriteNet+AttnX architecture to address these issues.

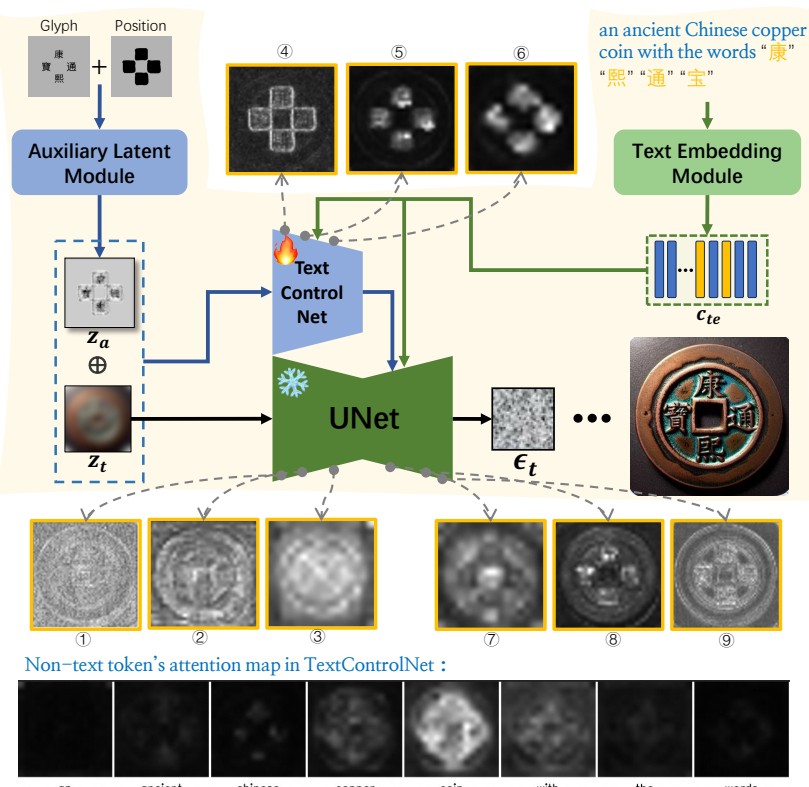

Figure 6: Analysis of the text generation process in AnyText.

## B    EXAMPLES OF FONT EXTRACTOR AND COLOR PICKER

In Fig. 7, we present examples of the extracted font image $e_f$ and text color $e_c$ obtained using the font extractor and color picker. Each set contains three images: the first is the training image, the second is the glyph image $l_g$ used in the Auxiliary Latent Module, which renders each line of text onto an image according to their positions using a glyph render. For display purposes, the color $e_c$ extracted by the color picker is applied to render text. Note that during training, $l_g$ does not include color information. Moreover, each text line is rendered using a randomly selected font to prevent the leakage of font style. This also brings the advantage that AnyText2 can choose any font file to generate text during inference, unlike AnyText which is limited to using the Arial Unicode font. The font image $e_f$ in the third image is extracted by the font extractor. To prevent the leakage of glyph, various transformations such as rotation, translation, scaling, and occlusion are applied.

## C    PREVENT WATERMARKS USING TRIGGER WORDS

Images containing text collected from the internet often come with numerous watermarks. According to AnyText Tuo et al. (2023), 25% of the Chinese data and 8% of the English data in the AnyWord-3M dataset are watermarked. They adopted a strategy of removing these watermarked images during the last two epochs of training, amounting to about 0.5 million images. We employed

Figure 7: Examples of the extracted font image and text color using the font extractor and color picker.

Table 4: Comparison with AnyText on watermark probabilities.

| watermark | Chinese | English |
|-----------|---------|---------|
| AnyText | 2.9% | 0.4% |
| AnyText2 | 1.8% | 0.7% |

a different approach that, based on the watermark probability provided in AnyWord-3M, labeled as *wm_score*, we added "no watermarks" to the captions with wm_score<0.5, and "with watermarks" for those with higher scores. During the inference, by adding the trigger words "no watermarks", watermarks can be effectively prevented. The comparison with AnyText on watermark probabilities is shown in Table 4.

## D  PARAMETER SIZE AND COMPUTATIONAL OVERHEAD OF ANYTEXT2

Our framework is implemented based on AnyText. Despite the addition of some modules, the total parameter sizes has only increased by 63.8M, as refered to Table 5. Moreover, due to the design of WriteNet that only performs inference once, the computational overhead is actually reduced. On a Tesla V100, the time taken to generate 4 images in FP16 has been reduced from 5.85s to 4.69s, resulting in a 19.8% improvement.

Table 5: The Comparison of the parameter sizes of modules between AnyText and AnyText2.

| Modules | AnyText | AnyText2 |
|---------|---------|----------|
| UNet | 859M | 859M |
| AttnX | - | 57M |
| VAE | 83.7M | 83.7M |
| CLIP Text Encoder | 123M | 123M |
| TextControlNet/WriteNet | 360M | 360M |
| Auxiliary Latent Module | 1.3M | 1.3M |
| Glyph Encoder | 4.6M | 4.6M |
| Position Encoder | - | 2.2M |
| Font Encoder | - | 4.6M |
| Color Encoder | - | 5K |
| Total | 1431.6M | 1495.4M |

# E EXAMPLES OF LONG AND SHORT CAPTIONS

From the examples presented in Fig. 8, it is evident that the short captions produced by BLIP-2 are very simplistic and may contain errors. In contrast, the long captions generated by QWen-VL not only provide a comprehensive description of the image details but also achieve a high level of accuracy, even accurately identifying the text within the images. We remove the quotation marks from these long captions and use them for training.

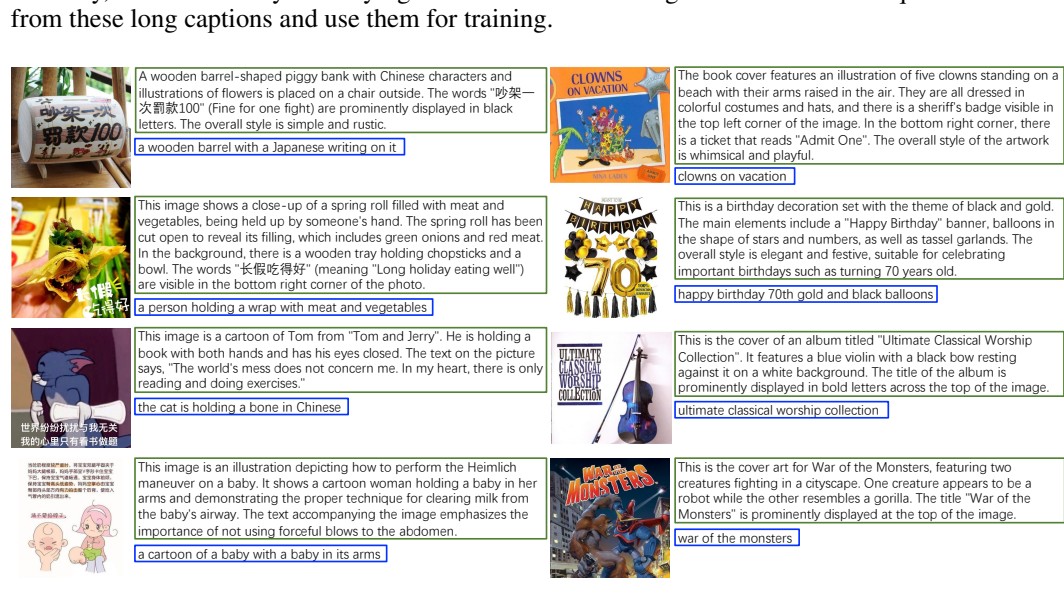

Figure 8: Exmaples of training images along with long and short captions by BLIP-2 and QWen-VL.

