# OpenReview forum: "ANYTEXT2: Visual Text Generation and Editing with Customizable Attributes"
_ICLR.cc/2025/Conference — ICLR 2025 Conference Withdrawn Submission_

### Official Review · Reviewer_RVwC · 2024-11-02

**Soundness:** 3
**Presentation:** 3
**Contribution:** 2
**Rating:** 5
**Confidence:** 4

**Summary:**

This paper achieves fine-grained control over different text attributes, including glyph, position, color, and font style, through multiple specialized encoders. Additionally, the decoupling of text and image generation via attention blocks enhances the realism of generated images. AnyText2 outperforms existing models, achieving superior accuracy and quality in text generation. Moreover, the use of extended captions has been validated to improve prompt-following capability and image realism. The proposed method not only achieves higher accuracy, enhanced realism, but also provides faster inference speed.

**Strengths:**

The text embedding module separately encodes the glyph, position, font and color attributes. Customizable font and color for each text line significantly enhance the visual appearance of the generated text.
2. The WRITENET+ATTNX architecture encodes text features and injects these intermediate features into the U-Net decoder via learnable attention layers, which decouples text generation from image generation and improves generation quality and inference speed.
3. By generating more complete and comprehensive descriptions of image details for training and evaluation, the model’s prompt-following capability is enhanced compared to using short captions.

**Weaknesses:**

1. To some extent, this work mainly builds on the structure of AnyText, with additional modules and architectural adjustments, which slightly limits its perceived novelty.
2. In the quantitative comparison between AnyText2 and other methods, using models trained on a long captions dataset to compare with previous methods that utilized short captions for training is not entirely fair. This approach obscures the contributions of the other modules in AnyText2.

**Questions:**

1. The model outputs complete long prompts. How are these prompts divided into text prompts and image prompts? Do the text prompts and image prompts require fixed templates, similar to the examples that start with 'Text are' as markers?
2. Could you explain the font encoder further? Which OCR model is used, and is it the same OCR model used for the glyph encoder?
3. In Sec3.1, ‘Notably, setting α = 0 and multiply the WriteNet output by 0 enables AnyText2 to generate images without text’，but as shown in the images, α = 0 means generating text only without rich background information. Why did you say that α = 0 generates images without text? How should this be understood?
4. Besides English and Chinese, is there an improvement in generation quality for other languages as well? Are there any illustrative examples of generated text images in multiple languages, such as Korean and Japanese, similar to those shown in AnyText?
5. The Glyph-ByT5 also achieves the text attributes customization as latest work, but why it is included in the qualitative results but not in the quantitative ones?
6. In WriteNet, is it reasonable to remove the noisy latent zt, and if so, does this affect the generation quality of the background content? Could you further explain the rationality behind this choice?

---

### Official Review · Reviewer_ZEqE · 2024-11-03

**Soundness:** 2
**Presentation:** 3
**Contribution:** 2
**Rating:** 5
**Confidence:** 3

**Summary:**

This paper extends AnyText work for more fine grained control of text attributes for visual text generation conditioned on text prompt and layout glyph mask. The papers proposes WriteNet+AttnX architecture that encodes text features and injects these intermediate features into the U-Net decoder via learnable attention layers. Text Embedding Module is used to employ multiple encoders to separately encode the glyph, position, font, and color of the text. Thorough evaluation was done on both Chinese and English cases.

**Strengths:**

- a new writenet+attx model, which is a controlnet like module to better decouple image generation from text generation. A self-attention and a cross-attention layer are inserted, denoted as AttnX layers to model text residual signals from the background, which are combined together with an output from each AttnX layer multiplied by a strength coefficient and combined with the output from the previous layer through a shortcut connection.
- Font, color and location are all separately encoded and conditioned.

**Weaknesses:**

- the controlnet style architecture is not new
- the color control example in Figure 4 still leaks into background objects
- the model is same as AnyText in most aspects, especially it’s conditioned on text layout mask which is not very general.

**Questions:**

- How well this model works without the glyph and position information in Fig 2. This can bring the model to the same setting as most T2I models.
- How’s the text rendering accuracy compared with most recent T2I models, such as Flux?

---

### Official Review · Reviewer_W9zF · 2024-11-04

**Soundness:** 3
**Presentation:** 3
**Contribution:** 2
**Rating:** 5
**Confidence:** 3

**Summary:**

This paper introduces AnyText2, advancing text-to-image generation by providing fine-grained control over text attributes (font, color, position) within images. The proposed WriteNet+AttnX architecture enhances realism and speeds generation, while the mixed Text Embedding Module improves text accuracy. Results validate AnyText2’s strong performance in realistic T2I text control.

**Strengths:**

1. The proposed WRITENET+ATTNX architecture enhances text generation capabilities.
2. Improved control over various text attributes is achieved through the introduction of a Text Embedding Module.
3. The method demonstrates better evaluation results compared to previous works, indicating improvements in text accuracy and image effects.
4. Figure 4 presents visually appealing results, showcasing the integration of visual text generation with style brush.

**Weaknesses:**

1. The proposed method does not differ significantly from previous works, particularly AnyText and Glyph-SDXL.
2. Figure 5 does not demonstrate substantial qualitative improvements over other models in comparison.
3. The paper lacks quantitative metrics for evaluating distinct text attributes, such as color, font, and position.
4. The text-heavy descriptions of different encoders are not entirely clear; additional figures would be beneficial to improve understanding.
5. A speed improvement over AnyText is mentioned in the abstract, this is not adequately discussed or compared with other methods in the main text.

**Questions:**

1. Does the method employ two separate models to support Chinese and English, or is a single model capable of handling both languages?
2. Why is AnyText not included in the comparison in Figure 5?
3. Is the model capable of generating large amounts of text within an image and what performance?

---

### Official Review · Reviewer_AZxq · 2024-11-04

**Soundness:** 4
**Presentation:** 3
**Contribution:** 3
**Rating:** 3
**Confidence:** 4

**Summary:**

The paper introduces AnyText2, a novel method for generating and editing multilingual text with customizable attributes within natural scene images. AnyText2 achieves precise control over text features, including glyphs, positions, fonts, and colors, through an efficient WriteNet+AttnX architecture and a Text Embedding Module. The method enhances the realism of generated images while improving the accuracy of text generation and allowing users to customize the font and color of each line of text. AnyText2 outperforms its predecessor, AnyText, on multiple evaluation metrics and plans to open-source code and models to foster the development of text generation technology.

**Strengths:**

1.  AnyText2's WriteNet+AttnX architecture effectively decouples text rendering tasks from image content generation while integrating them effectively through learnable attention layers (AttnX), improving inference speed and enhancing image realism.
2. By extracting font and color information from real images and using dedicated encoders for feature encoding, AnyText2 allows users to customize the font and color of each line of text, which is an innovative point in open-domain scenarios.

**Weaknesses:**

1. Many current generative models can already generate text quite well without such complex operations. Is the generative research presented in this paper unnecessary?
2. The color encoding will interfere with the model's ability. Why was it still added? In actual use, we do not need to control the color values very precisely.
3. Similarly, how does font encoding specify the same type of font when there are many fonts that are very similar, and some characters can even be interchanged between them. In such cases, what font do they each belong to? This is a question.
4. The current open-source optical character recognition (OCR) tools, such as DUGUANG, still have relatively low accuracy rates. For example, to my knowledge, the Chinese character recognition accuracy on the test set is only slightly above 80%, and it should be even lower on non-test sets. Therefore, using this OCR tool for testing may introduce excessive noise that could disrupt the test results. However, the results from the experiments in the paper appear to be quite regular. The authors should analyze this issue further, such as the relationship between the uncertainty of precision and the accuracy rate of the recognizer.

**Questions:**

1. The authors should provide quantitative experiments to demonstrate whether there is a performance decline in the model's general generative capabilities. If there is a performance decrease, then the necessity of the research focus of this paper should be considered more carefully.
2. How can you ensure that the CLIP text encoder has a good understanding of your rephrased longer prompts? What do you do in cases where the prompt exceeds the maximum length?
3. In Section 3.1, the authors need to provide quantitative changes in generation accuracy and FID scores during the variation of the strength coefficient to observe whether alpha has an impact on the accuracy of the generated content.
4. Please see weaknesses.

---

### Note · Authors · 2024-11-13

I have read and agree with the venue's withdrawal policy on behalf of myself and my co-authors.